# Effect of V Addition on Microstructure and Properties of Cu-1.6Ni-1.2Co-0.65Si Alloys

**Jinfeng Zou, Jianyi Cheng, Guangbo Feng, Jian Xie and Fangxin Yu ***

School of Materials Science and Engineering, Nanchang University, Nanchang 330031, China;
Jinfengzou1998@163.com (J.Z.); bigchengjianyi@163.com (J.C.); fenggb@email.ncu.edu.cn (G.F.);
xiejiancl@163.com (J.X.)
* Correspondence: yufangxin@ncu.edu.cn; Tel.: +86-189-7002-5670

**Abstract:** To obtain high strength and high electrical conductivity at the same time, the microstructure and properties of 0.2 wt.% V-added, 0.1 wt.% V-added and V-free Cu-1.6Ni-1.2Co-0.65Si(-V) alloys were investigated. We examined with electrical conductivity and hardness measurements, tensile test, optical microscope and transmission electron microscope (TEM). The results show that Cu-1.6Ni-1.2Co-0.65Si-0.1V alloy obtains excellent combination properties: electrical conductivity is 46.12% IACS, hardness is 293.88 Hv, and tensile strength is 782 MPa, which are produced by 65% cold rolling + aging at 500 °C for 480 min. The addition of vanadium (V) can accelerate the precipitation of solute atoms from the copper matrix, improve the hardness and electrical conductivity of Cu-1.6Ni-1.2Co-0.65Si alloys, and greatly accelerated the aging response. δ-$(Co,Ni)_2Si$ and β-$Ni_3Si$ phases are detected in Cu-1.6Ni-1.2Co-0.65Si-0.1V alloy. The Orowan mechanism and grain boundary strengthening play a major role in the yield strength strengthening due to δ-$(Co,Ni)_2Si$ phase.

**Keywords:** Cu-Ni-Co-Si alloy; precipitate; aging; microstructure; property

## 1. Introduction

With the rapid development of the electronics industry, copper alloys are widely used for contact wires and lead frames due to their good electrical conductivity, good mechanical properties and low cost [1–7], such as Cu-Cr [8], Cu-Ni-Sn [9], Cu-Ni-Si [10] and Cu-Ti [11] alloys. It is difficult for copper alloys to obtain high strength and electrical conductivity at the same time. Cu-Ni-Si alloy gets much attention because of a balance between high tensile strength and high electrical conductivity. Cu-Ni-Si alloy is generally processed through a series of mechanical processes to form nanoscale precipitates of δ-$Ni_2Si$ and β-$Ni_3Si$ in the copper matrix [12], which effectively prevents dislocations and grain boundary motion [13]. The high content of Ni and Si in Cu-Ni-Si alloy can significantly increase the tensile strength of the alloy, while the elongation decreases slightly [14].

In the modern electronics industry, many methods of alloying have been proposed as devices tend to be miniaturized. Many studies have tested the addition of other elements in Cu-Ni-Si alloys in order to provide new materials for the electronic industry with both high strength and high electrical conductivity, such as Al, Ti, P, Mg, Cr, Zr, etc., indicating a significant improvement in alloy properties [3,14–18]. Al-alloying can increase anti-stress relaxation resistance. Ti-alloying can improve the elongation of samples and refine grain. The addition of Mg can obtain high tensile strength due to Mg-atom-drag effect on dislocation motion. Cr- and Zr-alloying can enhance conductivity by formation of $Cr_3Si$ and $Ni_2SiZr$, but the addition of Zr reduces the mechanical strength. Cu-Ni-Si-Co alloy is a third-generation lead frame material newly developed by Olin Company of the United States, which has better comprehensive properties than Cu-Ni-Si alloy, because Co can suppress spinodal decomposition and enhance electrical conductivity and mechanical properties [19,20]. The required

content of Ni is less than 3% to obtain a high electrical conductivity [21]. Moreover, the addition of vanadium in Cu-Ni-Si alloy can increase the hardness and electrical conductivity of the alloy due to the influence of vanadium atoms on dislocation motion [22].

Yang et al. found that elongation decreases with cold working and found that high cold rolling is beneficial to the formation of twin and copper textures. The hardness of the alloy increases by cold rolling because of high density of dislocations generated by cold rolling [13,23,24]. Li et al. found a combination of medium tensile strength and high electrical conductivity obtain through two-step cryorolling and aging, and found that the pre-deformation accelerates the aging process, promotes precipitation, and increases strength [25,26]. We aim to achieve an optimum combination of super-high electrical conductivity and high tensile strength to set different mechanical processes for treatment as shown in Table 1.

**Table 1.** Treatment processes.

| Treatment | Process | Treatment Process |
|---|---|---|
| Solution treatment | Process 1 | Homogenizing at 850 °C for 10 min and cold rolling into 1.56 mm and solution treatment at 950 °C for 10min and then aging at 500 °C from 0 to 480 min. |
| Cold rolling treatment | Process 2 | Solution treatment at 950 °C for 30 min and cold rolling into 1.40 mm and aging at 500 °C from 0 to 480 min. |
| Two-step aging treatment | Process 3 | Solution treatment at 950 °C for 30 min and pre-aging at 500 °C for 2 h and cold rolling into 1.60 mm; and aging at 350 °C from 0 to 9 h (final aging) |

This paper focuses on the effects of different contents of vanadium and different processes of Cu-1.6Ni-1.2Co-0.65Si(-V) alloys. Based on the experimental results, we related the electrical and mechanical properties achieved after different heat treatments to careful microstructure analyses.

## 2. Materials and Methods

Cu-1.6 wt.% Ni-1.2 wt.% Co-0.65 wt.% Si alloys with the addition of V were prepared by a vacuum induction furnace, the compositions of which were pure copper, pure nickel, pure cobalt, pure silicon, and Ni-50%V master alloys. The nominal composition of the Cu-1.6 wt.% Ni-1.2 wt.% Co-0.65 wt.% Si alloy is shown in Table 2. The temperature of the alloy to be molten was cooled to 1200 °C. The ingot was cast in a graphite mold with dimensions of 120 mm × 20 mm × 72 mm, homogenized at 950 °C for 24 h, and removed the macroscopic surface defects by machining, and hot rolling by 80% reduction at 850 °C, then treated by different treatment process shown in Table 1.

**Table 2.** The nominal composition of the Cu-Ni-Co-Si alloys.

| Specimen Designation | Content (wt.%) | | | | |
|---|---|---|---|---|---|
| | V | Ni | Co | Si | Cu |
| Cu-Ni-Co-Si | - | 1.60 | 1.19 | 0.65 | Balance |
| Cu-Ni-Co-Si-0.1V | 0.10 | 1.60 | 1.20 | 0.65 | Balance |
| Cu-Ni-Co-Si-0.2V | 0.21 | 1.60 | 1.18 | 0.65 | Balance |

The hardness was measured with the load of 0.2 kg and holding time of 10 s by MH-50 type microhardness tester (HanJi, Shenzhen, China). The electrical conductivity was measured by a FQR-7501 electrical instrument (SiTe, Taiwan) at 25 °C. Tensile testing used an INSTRON 2380 tester machine (INSTRON, Norwood, MA, USA), with a strain rate of 1 mm/min. The metallographic structure of the sample was etched with a solution of 8 g (5 g $FeCl_3$ + 25 mL HCl + 100 mL $H_2O$), and observed with a XJZ-6Z optical microscope (JiangNan, Nanjing, China). Transmission electron microscopy (TEM) samples were prepared by electrolytic double spray thinning. TEM detection, SADP

selection electron diffraction analysis and EDS pattern analysis were performed using FEI Talos F200X microscope (FEI, Hillsboro, OR, USA) equipped with EDS, with an operating voltage of 200 kV.

## 3. Results

### 3.1. Electrical Conductivity and Hardness

Figure 1 shows the electrical conductivity and hardness of the solution treatment samples with 0.1 wt.% V, 0.2 wt.% V and without V (process 1). The samples were aged at 500 °C for different time. The electrical conductivity increased quickly at the beginning of aging time and gradually increased during the aging process. Although the samples were aged for a long time, the electrical conductivity did not reach the peak value. The electrical conductivity values of the samples with the addition of 0.2 wt.% V, 0.1 wt.% V and V-free were 51.64% IACS, 49.83% IACS, and 42.24% IACS at peak aging time. For relatively long aging time at 500 °C, the samples with the addition of 0.2 wt.% V showed the better electrical conductivity than the samples with the addition of 0.1 wt.% V and V-free. The hardness of the samples with the addition of 0.2 wt.% V, 0.1 wt.% V and V-free were 210.15 Hv, 222 Hv, and 217.64 Hv at peak aging time. After reaching the peak value, the hardness decreased slightly. Figure 2 shows the micrographs of Cu-1.6Ni-1.2Co-0.65Si(-V) alloys aged at 500 °C for 120 min after solution treatment. The grains of sample with the addition of 0.1 wt.% V were slightly refined. In addition, thermal twins could be observed in three samples by solution treatment.

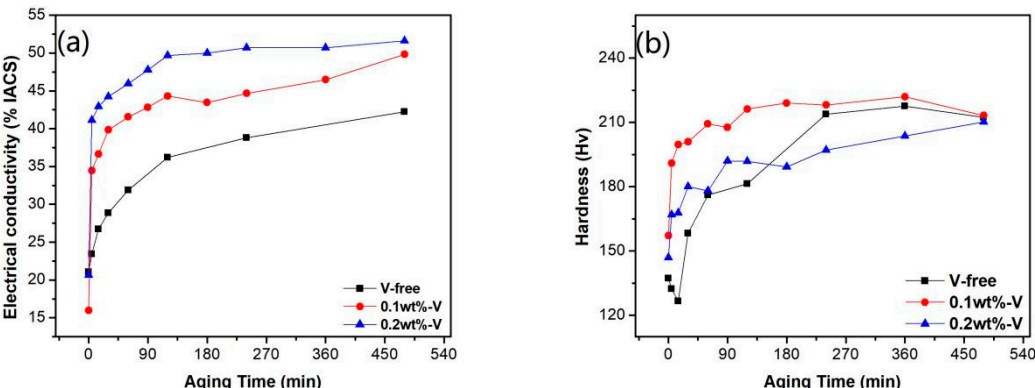

**Figure 1.** (**a**) Electrical conductivity and (**b**) hardness of the solution treatment samples with 0.1 wt.% V, 0.2 wt.% V and without V.

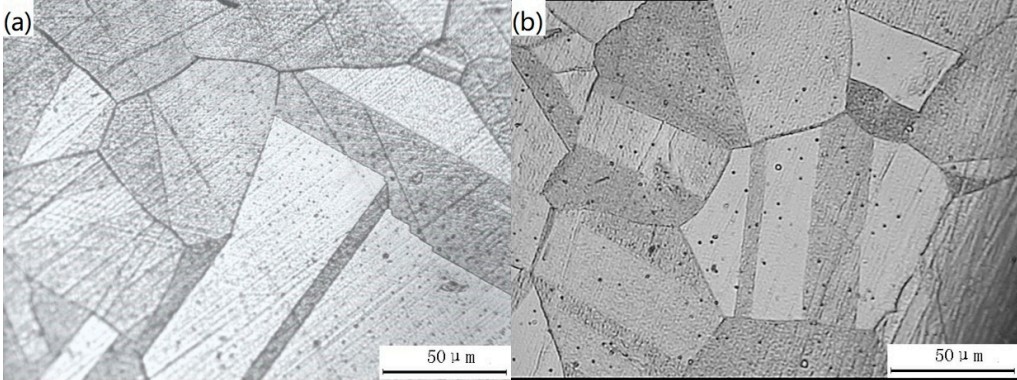

**Figure 2.** The micrographs of Cu-1.6Ni-1.2Co-0.65Si(-V) alloys aged at 500 °C for 120 min after solution treatment without (**a**) V and with (**b**) 0.1 wt.% V.

Figure 3 shows electrical conductivity and hardness of cold-rolled samples with 0.1 wt.% V, 0.2 wt.% V and without V (process 2). The electrical conductivity gradually increased during the aging process. The variation trend of electrical conductivity was similar to Figure 1a. Although the samples

were aged for a long time, the electrical conductivity did not reach the peak value except for the samples with 0.1 wt.% V, and the other two groups did not show significant peak values. The electrical conductivity values of the samples with the addition of 0.2 wt.% V, 0.1 wt.% V and V-free were 51.64% IACS, 49.83% IACS, and 42.24% IACS at peak aging time. The hardness of the samples increased quickly at the beginning of aging time, and decreased after reaching the peak-aged values because of over-aging. The values of hardness approached 290.55 Hv, 293.88 Hv and 253.88 Hv for peak-aged samples with the addition of 0.2 wt.% V, 0.1 wt.% V and V-free. The time to reach the peak-aged hardness was shortened, and hardness decreased slightly during over-aging due to the addition of vanadium. Figure 4 shows the micrographs of Cu-1.6Ni-1.2Co-0.65Si(-V) alloys aged at 500 °C for 240 min after cold rolling. The grains were elongated due to cold rolling and the microstructure was highly oriented, but the microstructure was heterogenous due to cold rolling.

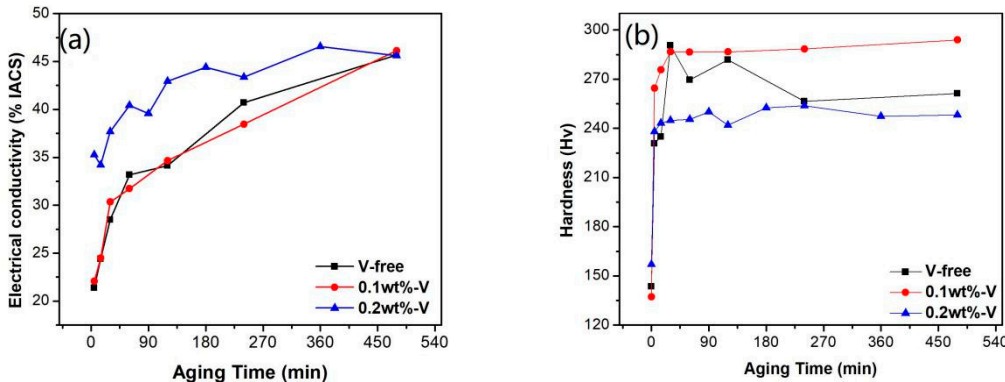

**Figure 3.** (**a**) Electrical conductivity and (**b**) hardness of the cold-rolled samples with 0.1 wt.%, 0.2 wt.% V and without V.

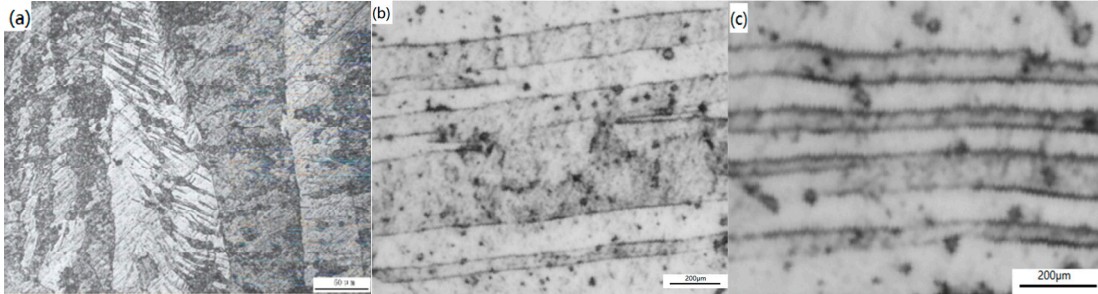

**Figure 4.** The micrographs of Cu-1.6Ni-1.2Co-0.65Si(-V) alloys aged at 500 °C for 240 min after cold rolling without (**a**) V, and with (**b**) 0.1 wt.% V and (**c**) 0.2 wt.% V.

Figure 5 shows the effect of electrical conductivity and hardness of two-step aged samples with the aging time. Although the samples were aged for a long time, the electrical conductivity was still increasing. The electrical conductivity values of the samples with the addition of 0.2 wt.% V, 0.1 wt.% V and V-free were 42.7% IACS, 46.55% IACS and 40.09% IACS at peak aging time. The hardness of the samples with 0.1 wt.% V-added aged for 120 min obtained peak-aged values of 271 Hv. The hardness of the samples decreased slightly to 268.27 Hv when the aging time increased to 240 min. Therefore, the two-step aging process could rapidly improve properties of the alloy [13].

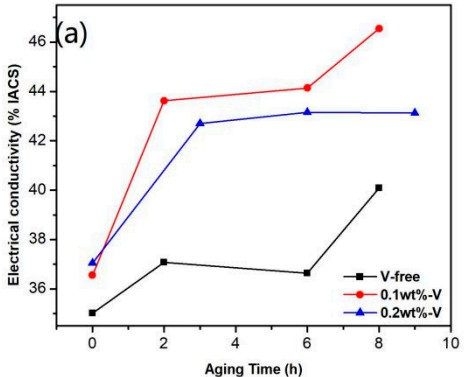
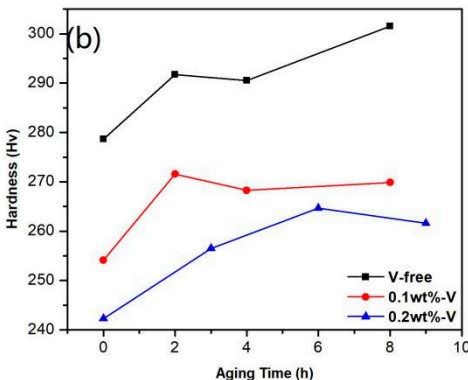

**Figure 5.** (**a**) Electrical conductivity and (**b**) hardness of two-step aged samples with 0.1 wt.%, 0.2 wt.% V and without V.

### 3.2. Mechanical Properties

The peak-aged properties of Cu-1.6Ni-1.2Co-0.65Si(-V) alloys are shown in Table 3. The samples aged at 500 °C for 480 min with the addition of 0.1 wt.% V by Process 2 has excellent combination properties: electrical conductivity was 46.12% IACS, hardness was 293.88 Hv, yield strength was 721.76 MPa, elongation was 6.5%, and tensile strength was 782.20 MPa.

**Table 3.** Peak-aged properties of the alloys.

| Process | Addition (wt.%) | Time (min) | Electrical Conductivity (%IACS) | Hardness (Hv) | Yield Strength (MPa) | Elongation (%) | Tensile Strength (MPa) |
|---|---|---|---|---|---|---|---|
| 1 | 0.1 | 240 | 44.66 | 218.15 | 533.15 | 6.0 | 687.69 |
| 1 | 0.1 | 360 | 46.47 | 222.00 | 547.23 | 5.6 | 687.31 |
| 1 | 0.2 | 360 | 50.69 | 203.64 | 536.23 | 9.8 | 711.59 |
| 1 | 0.2 | 480 | 51.64 | 210.15 | 565.37 | 9.4 | 726.52 |
| 2 | 0.1 | 240 | 38.45 | 288.45 | 684.83 | 4.2 | 752.18 |
| 2 | 0.1 | 480 | 46.12 | 293.88 | 721.76 | 6.5 | 782.20 |
| 2 | 0.2 | 180 | 44.40 | 252.52 | 672.60 | 3.3 | 733.80 |
| 2 | 0.2 | 240 | 43.36 | 253.88 | 720.57 | 8.1 | 769.05 |

### 3.3. Microstructure of Samples Treated by Solution Treatment (Process 1)

We examined the microstructure, precipitation behavior and morphology of samples treated by solution treatment. Figure 6 shows the TEM micrographs and HRTEM image of the samples with the addition of 0.1 wt.% V aged at 500 °C for 240 min. Figure 6a shows Ni and Si solute atoms are precipitated from supersaturated solid solution. The precipitates were evenly distributed in the field of view and observed the dislocation tangles in the sample. Figure 6b gives the selected-area diffraction patterns (SADP) of the alloy. SADP shows δ-(Co,Ni)$_2$Si and β-Ni$_3$Si phase are the contributor of diffraction spots. These were consistent with the results reported in reference [27–29]. Figure 6c shows that the HRTEM images with the dimension of the disk-like precipitates is 12–13 nm. Figure 7 shows the TEM images of disk-like precipitates in the Cu-1.6Ni-1.2Co-0.65Si-0.1V alloy aged at 500 °C for 240 min. After aging for 240 min, δ-(Co,Ni)$_2$Si precipitates were observed along the grain boundaries, but not in the copper matrix.

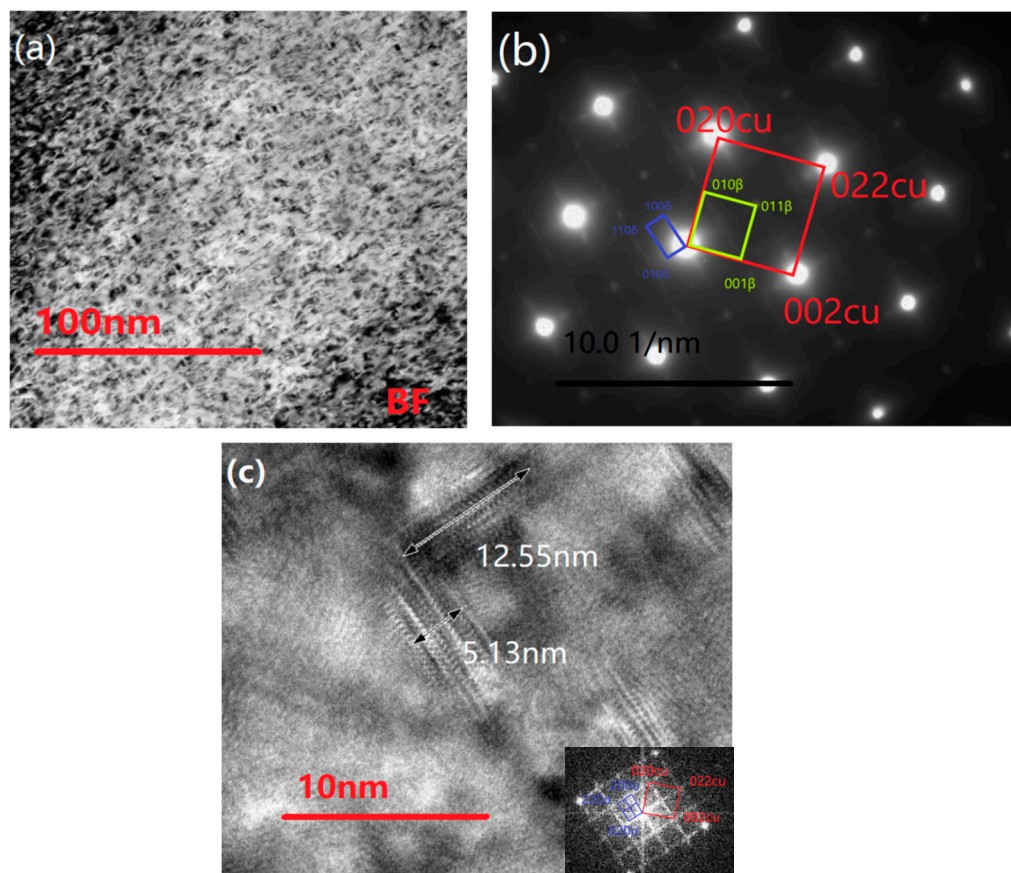

**Figure 6.** Transmission electron microscope (TEM) micrographs and high Resolution Transmission electron microscope (HRTEM) image of the samples with the addition of 0.1 wt.% V aged at 500 °C for 240 min: (**a**) bright-field TEM micrograph of precipitates; (**b**) SADP of (**a**) with the beam along zone axis: [001]Cu; and (**c**) HRTEM image and fast fourier transformation (FFT) patterns of δ-(Co,Ni)$_2$Si precipitates.

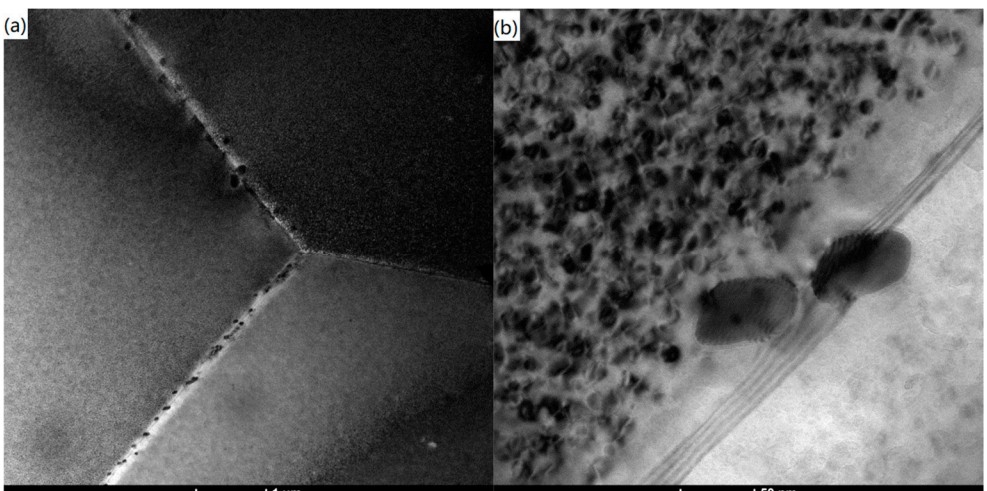

**Figure 7.** TEM micrographs of disk-like precipitates in the alloy with the addition of 0.1 wt.% V aged at 500 °C for 240 min: (**a**) the precipitated Ni particles. (**b**) the relatively fine δ-(Co,Ni)$_2$Si phase.

Figure 8 shows the HAADF micrograph, composition of disk-like precipitates and elemental mapping images from samples aged at 500 °C for 16 h. According to elemental mapping images, the nanoscale particles were observed. The nanoscale precipitates were rich in Ni, Co and Si elements.

V elements were evenly distributed in copper matrix. Since the atomic radii of Ni and Co were similar, the δ-phase consists of δ-Ni$_2$Si and δ-Co$_2$Si [27]. Two phases were difficult to distinguish because of the same structure and similar lattice constants. According to the Energy Dispersive Spectrum (EDS) results, the chemical composition of the particles in Area #2 (labeled in Figure 8a) is (at.%): 4.30Ni, 8.70Co, 10.52Si, 0.25V and 76.23Cu. The (Ni,Co)/Si ratio is about 1.2. The nanoscale particles were close to δ-(Co,Ni)$_2$Si phase. This was consistent with the selected-area diffraction patterns (SADP) results.

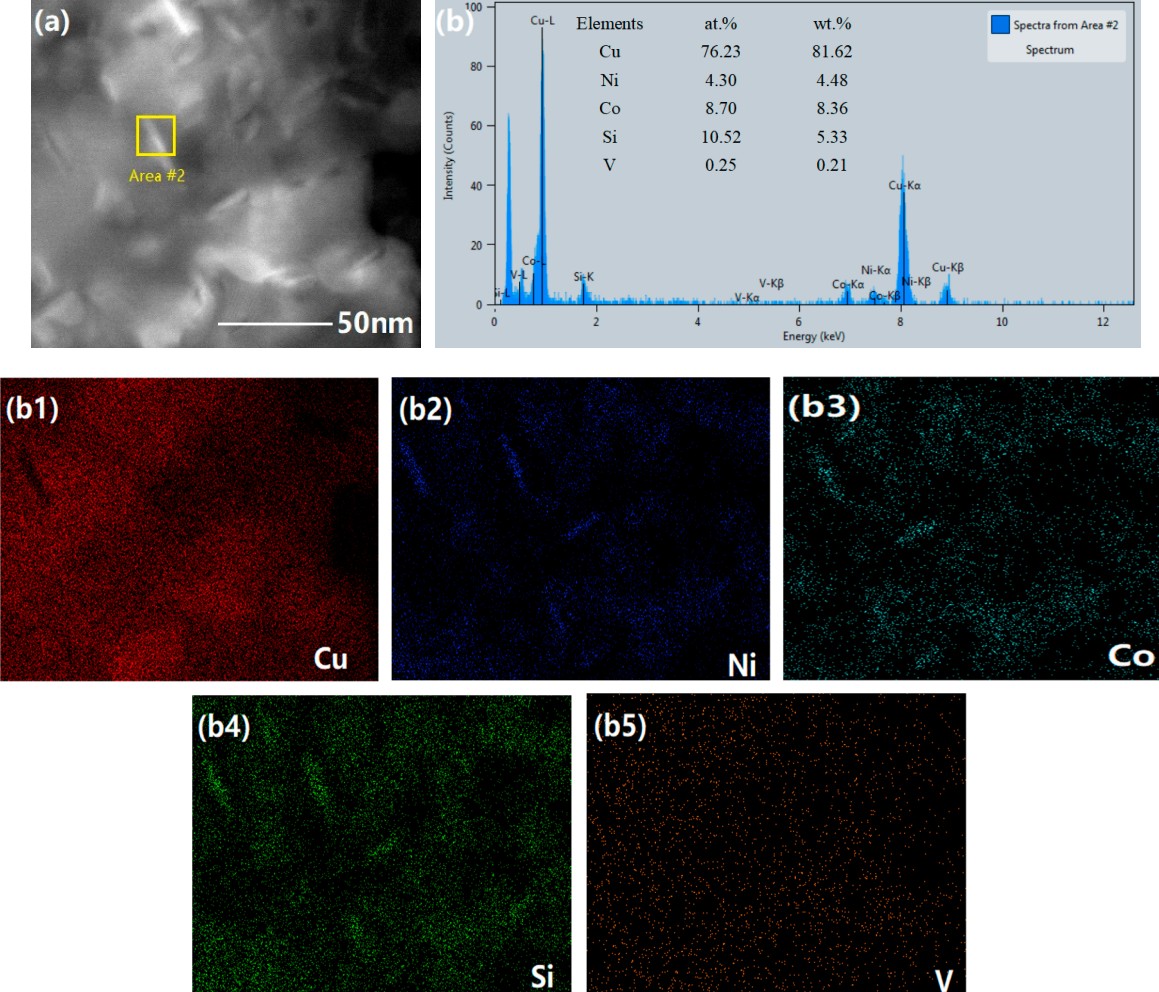

**Figure 8.** (**a**) High-angle annular dark field (HAADF) micrograph, (**b**) composition of disk-like precipitates and elemental mapping images from the samples aged at 500 °C for 16 h (0.1 wt.% V-added): (**b1**–**b5**) Cu, Ni, Co, Si and V.

Figure 9 shows the TEM micrographs and HRTEM image of the samples with the addition of 0.1 wt.% V aged at 500 °C for 16 h. Figure 9a shows that the δ-(Co,Ni)$_2$Si and β-Ni$_3$Si particles are significantly coarsened and the dislocation entanglement still exist. The dislocation density obviously decreased because the precipitation nucleation required absorbing energy. Figure 9b shows that the δ-(Co,Ni)$_2$Si precipitates have mutually perpendicular growth directions, and they significantly increase in size (Figure 6a). The dimensions of the precipitates were 16–17 nm (Figure 9c), which is bigger than in Figure 6c (aged for 240 min). The dimensions of the precipitates increased slightly during the aging process. The SADP image had no significant changes. The spots of copper matrix, δ-(Co,Ni)$_2$Si and β-Ni$_3$Si precipitates could be detected.

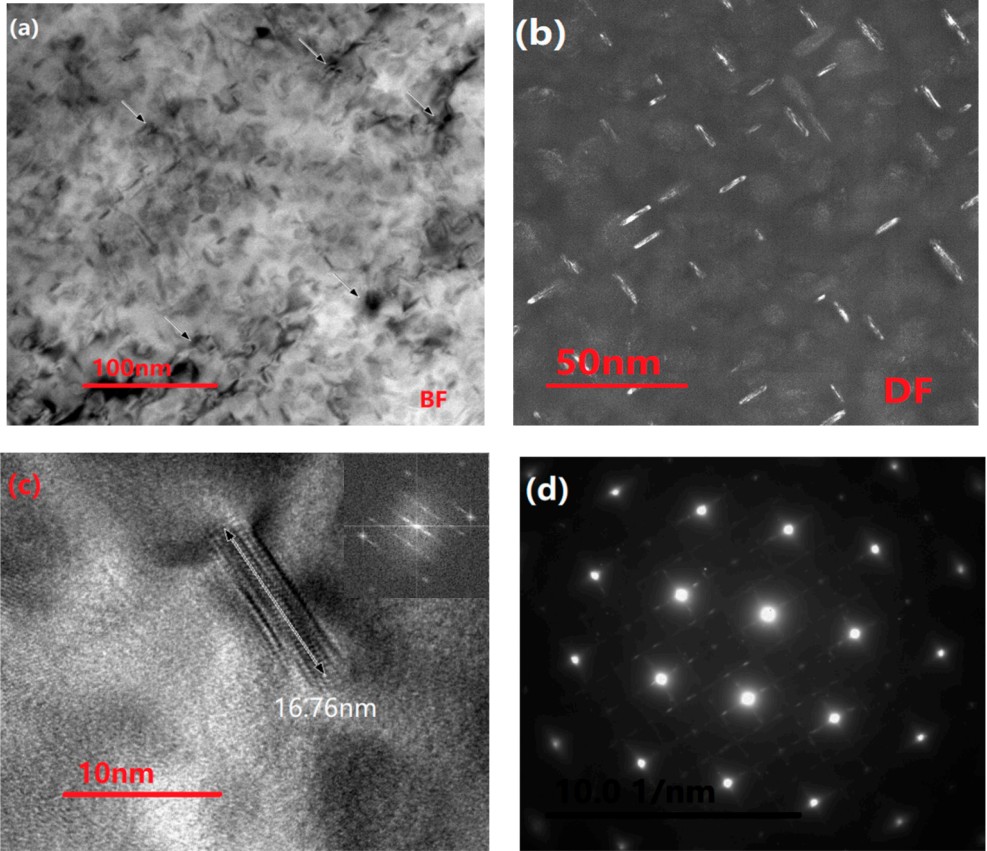

**Figure 9.** TEM micrographs and HRTEM image of the samples with the addition of 0.1 wt.% V aged at 500 °C for 16 h: (**a**) bright-field TEM micrograph of precipitates; (**b**) dark-field TEM micrograph of precipitates; (**c**) HRTEM image and FFT patterns of δ-(Co,Ni)$_2$Si precipitates; and (**d**) [001]Cu SADP.

Figure 10 shows the TEM micrographs and of the HRTEM image of the samples with the addition of 0.2 wt.% V aged at 500 °C for 120 min. Figure 10a shows that more particles are precipitated in the field of view are δ-(Co,Ni)$_2$Si and β-Ni$_3$Si phases. Figure 10b shows that the precipitates are more uniformly distributed in the field of view, than in Figure 9b. The dimensions of the precipitates are about 21–22 nm (Figure 10c). Although the aging time was shorter, the dimensions of the precipitates were larger. This means that the addition of a small amount of vanadium could accelerate the precipitation of solute atoms from the matrix. The large coarsened precipitate particles may disadvantageously influence the hardness.

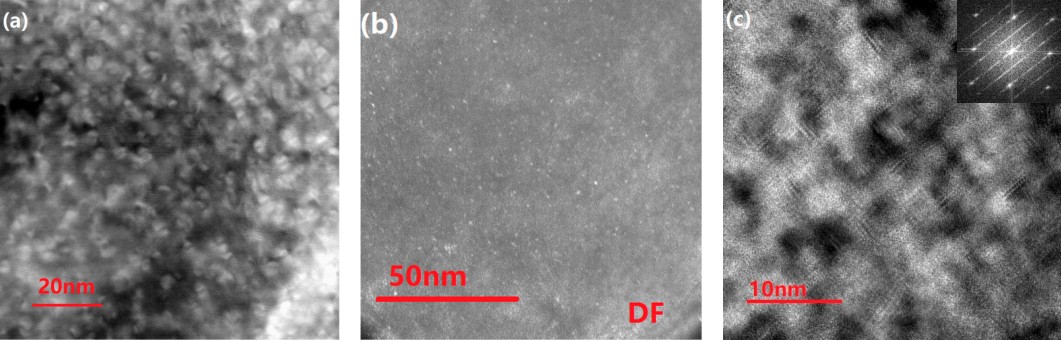

**Figure 10.** TEM micrographs and HRTEM image of the samples with the addition of 0.2 wt.% V aged at 500 °C for 120 min: (**a**) HAADF micrograph; (**b**) dark-field TEM micrograph of precipitates; and (**c**) HRTEM image and FFT patterns of δ-(Co,Ni)$_2$Si precipitates.

## 4. Discussion

### 4.1. Electrical Conductivity

According to Matthiessen's rule [30], the electrical resistance resistivity can be expressed by:

$$\rho = \rho_T + \rho_S + \rho_P + \rho_D \tag{1}$$

where $\rho_T$ is the basic electrical resistivity of metal and changes with temperature, $\rho_S$ is electrical resistivity associated with solute atoms, $\rho_P$ is electrical resistivity associated with precipitates, $\rho_D$ is electrical resistivity associated with defects. Generally, $\rho_D << \rho_S$ [31]. $\rho_T$ is an invariant because all electrical conductivity measurements were examined at 25 °C. Therefore, the $\rho_S$ is the major contributor of electrical conductivity of alloys. The electrical conductivity values of samples with the addition of 0.1 wt.% V is 46.12% IACS treated by cold rolling. This is attributed to the fact that the vanadium can accelerate the precipitation of solute atoms from the matrix and cold rolling process can promote solute atom precipitation. The precipitates in Figure 10c are much bigger than in Figures 6c and 9c. This explains that the addition of 0.2 wt.% V can make the electrical conductivity more excellent.

Table 4 shows the electrical conductivity of Cu-1.6Ni-1.2Co-0.65Si(-V) alloys aged for 5 min in different process. V is a refractory metal and V could prepare high melting point intermetallic compounds. V elements distributed at the grain boundary and may provide nucleation sites for the precipitated phase. Therefore, V could contribute to promote δ-(Co,Ni)$_2$Si and β-Ni$_3$Si precipitation, and causes the electrical conductivity to increases with V content. The electrical conductivity of the alloys decreases due to the cold rolling process. Because a high density of dislocations is introduced by cold rolling, and causes $\rho_D$ to increase. The precipitates were formed during pre-aging. The two-step aging treatment contributed to reducing the content of Ni and Si in the solid solution and improve the electrical conductivity.

**Table 4.** Electrical conductivity (% IACS) of Cu-1.6Ni-1.2Co-0.65Si(-V) alloys aged for 5 min in different process.

| Process | Cu-Ni-Co-Si | Cu-Ni-Co-Si-0.1V | Cu-Ni-Co-Si-0.2V |
|---|---|---|---|
| Solution treatment | 23.45 | 34.48 | 41.12 |
| Cold rolling | 21.38 | 22.07 | 35.26 |
| Two-step aging | 35.02 | 36.55 | 37.04 |

### 4.2. Property Comparison

Figure 11 shows the properties of various alloys with the designed alloy. Cu-Cr-Zr alloys show super-high electrical conductivity (84.7% IACS) and low tensile strength. Cu-Ni-Si system alloys show a medium tensile strength and high electrical conductivity, such as Cu-Ni-Si-Ti, Cu-Ni-Si-Cr-Zr, Cu-Ni-Si-Cr, Cu-Ni-Si-Mg and Cu-Ni-Si-Zr alloys. Cu-Ni-Zn, Cu-Ni-Sn and Cu-Ti alloys, designed by previous researchers, showed tensile strength higher than 1000 MPa, but their electrical conductivity was lower than 12% IACS. The Cu-Be alloys show excellent tensile strength and medium electrical conductivity, but anti-stress relaxation resistance is low. The Cu-1.6Ni-1.2Co-0.65Si-0.1V alloy designed in this work possesses a super-high electrical conductivity under the premise of high tensile strength, because the deformation in the cold rolling process is large and the addition of 0.1 wt.% V results in fine grain strengthening. However, the effect of fine grain strengthening is limited, and is much worse than the Orowan strengthening. This work provides a choice for the next generation of flexible conductive materials.

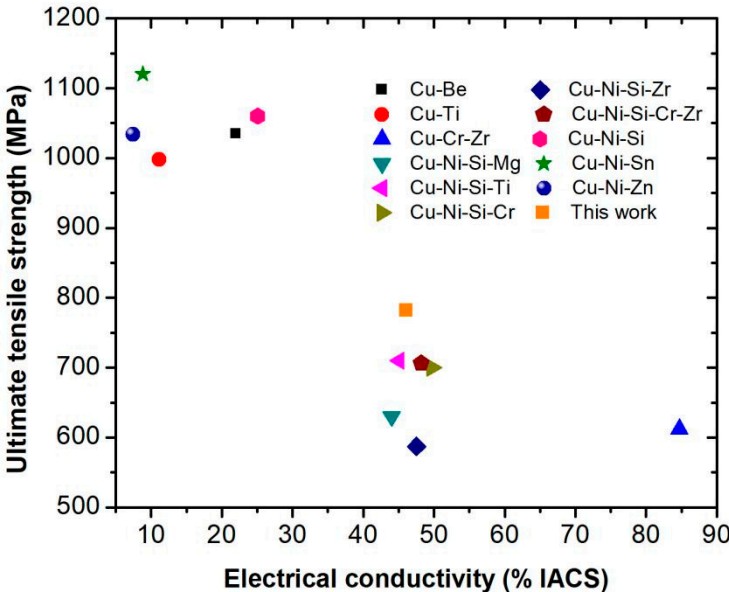

**Figure 11.** Properties of various alloys with the designed Cu-1.6Ni-1.2Co-0.65Si-0.1V alloy [8,18,32–40].

### 4.3. Mechanical Properties

The yield strength strengthening mechanisms of the designed alloy can be divided into: (1) grain boundary strengthening, (2) solid solution strengthening, (3) precipitation strengthening, (4) phase transformation strengthening, (5) order strengthening. Among these, precipitation strengthening and grain boundary strengthening play a major role in these alloys.

#### 4.3.1. Precipitation Strengthening

We assume that all precipitates are δ-(Co,Ni)$_2$Si for research convenience. The alloys with δ-(Co,Ni)$_2$Si are controlled by Orowan strengthening [18]. The yield stress is independent of the nature of the precipitate, only depending on the particle spacing λ, and is inversely proportional to λ [35]. It can be expressed by [41]:

$$\lambda = r_0 \left[ \sqrt{\frac{2\pi}{3f}} - 1.63 \right] \tag{2}$$

where $r_0$ and $f$ stand for the radius and volume fraction of precipitates. δ-(Co,Ni)$_2$Si phase gradually forms during aging process. Therefore, the volume fraction of the precipitate decreases. According to Equation (2), λ is inversely proportional to $f$ and yield stress increases. When the volume fraction of the precipitates is constant, the $r_0$ increases due to the coarsening of the precipitates. λ increases when $r_0$ decreases, and then yield stress decreases. Therefore, we need to determine the appropriate aging time, aging process and chemical composition to control the coarsening of the precipitate.

#### 4.3.2. Grain Boundary Strengthening

Generally, the Hall-Petch equation is used to explain grain boundary strengthening. According to the Hall-Petch equation [42], the relationship between the yield stress of a polycrystal and the average grain diameter can be expressed by:

$$\sigma_S = \sigma_0 + kd^{-\frac{1}{2}} \tag{3}$$

where $\sigma_0$ is the lattice frictional stress and is a constant, $d$ is the average diameter of the grains and k is a constant. The addition of 0.1 wt.% V has the effect of refining grains as shown Figure 2. Therefore, the diameter of the grains with the addition of 0.1 wt.% V is relatively small. According to Equation (3), $\sigma_s$ is inversely proportional to $d$. Therefore, the yield strength of the samples with the addition of 0.1 wt.% V is larger than that without V. The precipitates of samples with the addition of 0.2 wt.% V are

obviously coarsening, as shown Figures 6c, 9c and 10c. This causes the nucleation rate of the crystal to decrease, resulting in a decrease in grain boundaries, and then the yield stress decreases. Therefore, the sample with the addition of 0.1 wt.% V has the largest yield strength.

*4.4. Hardness*

Hardness is positively correlated with ultimate tensile strength. The increase in dislocation density due to plastic deformation leads to an increase in the interaction between dislocations. A large number of obstacles such as entanglement and fixed dislocations are formed, which increases the resistance of other dislocation motions. Therefore, the hardness is greatly improved. Since the amount of rolling deformation set by each researcher is not necessarily the same, we discuss the hardness value under solution treatment here in order to make this work more general.

Table 5 shows the hardness of Cu-1.6Ni-1.2Co-0.65Si(-V) alloys aged for 5 min in different process. The solution treatment mainly consists of three stages: recovery, recrystallization and grain growth. The variation of hardness mainly occurs in the last two stages. The effect of V on hardness is similar to the effect of grain boundary strengthening on yield strength.

**Table 5.** Hardness (Hv) of Cu-1.6Ni-1.2Co-0.65Si(-V) alloys aged for 5 min in different process.

| Process | Cu-Ni-Co-Si | Cu-Ni-Co-Si-0.1V | Cu-Ni-Co-Si-0.2V |
|---|---|---|---|
| Solution treatment | 132.28 | 191.02 | 166.85 |
| Cold rolling | 230.83 | 264.55 | 237.90 |
| Two-step aging | 278.67 | 254.13 | 242.26 |

## 5. Conclusions

(1) Cu-1.6Ni-1.2Co-0.65Si-0.1V alloy obtained excellent combination properties: electrical conductivity was 46.12% IACS, hardness was 293.88 Hv, yield strength was 721 MPa, tensile strength was 782 MPa, and elongation was 6.5%, which were produced by 65% cold rolling + aging at 500 °C for 480 min.

(2) The addition of vanadium could accelerate the precipitation of solute atoms from the matrix, improve electrical conductivity of Cu-1.6Ni-1.2Co-0.65Si alloys, and greatly accelerated the aging response.

(3) $\delta$-(Co,Ni)$_2$Si and $\beta$-Ni$_3$Si phases were detected in Cu-1.6Ni-1.2Co-0.65Si-0.1V alloy. The Orowan mechanism and grain boundary strengthening played a major role in the yield strength strengthening due to $\delta$-(Co,Ni)$_2$Si phase.

**Author Contributions:** Conceptualization, J.C. and F.Y.; Data curation, J.Z., G.F. and J.X.; Formal analysis, J.Z. and G.F.; Funding acquisition, J.C.; Investigation, J.Z.; Methodology, F.Y.; Project administration, F.Y.; Resources, J.C.; Supervision, F.Y.; Validation, J.Z.; Writing—original draft, J.Z. and J.X.; Writing—review & editing, F.Y.

**Funding:** This work was supported by National key research and development program of China (2016YFB0301302), and National Natural Science Foundation of China (51661022).

**Conflicts of Interest:** The authors declare no conflict of interest.

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
