# Peer review of "Effect of V Addition on Microstructure and Properties of Cu-1.6Ni-1.2Co-0.65Si Alloys"

_metals, doi:10.3390/met9060679_

Reviewer 1 Report

The manuscript subject is too interesting and the authors did good efforts in this work. However, the manuscript should be revised carefully for the following major revisions:

The introduction should be reorganized and modified to more clarify basic background of alloys studied, its engineering or industrial application, why have been selected and the main objective of work,.

What about the other published works relating to this subject. The authors should add more texts that clearly indicate their point of view with respect to literatures.

The methodology is not clear enough to make the readers of Metals journal can follow this work. More details are required to be added into this section. Schematic figure could be helpful.

Fiqure 4 is not qualitative. The authors have to replace it with high quality image.

Figure 8, the shown precipitates are not going well with the direction indicated in the diffraction pattern. Please verify that and supply the good diffraction pattern directions that identify the type of precipitates formed.

Figures 10.a, c need to replace with high quality one. The authors should improve the image quality to better clear their point of view.

Discussions are not enough. The analysis and interpretation of your results should be improved. Speculative or descriptive discussions are not helpful to reach at a better understanding of the main aim of this work (the point of research). The authors are encouraged to add more explanations with respect to literature. So, add some new references to this section with more explanations.

Conclusions are good but need to be more specific and having the most new points drawn from this work. The conclusions that may add to literature as new points.

Author Response

Dear reviewers:

Thank you very much for your careful review and constructive suggestions with regard to our manuscript “Effect of V Addition on Microstructure and Properties of Cu-1.6Ni-1.2Co-0.65Si Alloys” (ID:metals-520883). Those comments are helpful for authors to revise and improve our paper. We have studied comments carefully and tried our best to revise and improve the manuscript and made great changes in the manuscript according to the referees′ good comments. Revised portion is marked in yellow in the paper. The main corrections in the paper and the responds to the reviewer’s comments are as flowing. We appreciate for Reviewers’ warm work earnestly, and hope that the corrections will meet with approval. Please feel free to contact us with any questions and we are looking forward to your consideration. The main corrections in the paper and the responds to the reviewer’s comments are as flowing:

Responds to the reviewer’s comments:

REVIEWER #1:The introduction should be reorganized and modified to more clarify basic background of alloys studied, its engineering or industrial application, why have been selected and the main objective of work,.

Responds:We found the referee’s comments most helpful and have revised the manuscript.

We have revised our introduction, and the revised part in introduction is following:In the modern electronic industry, many methods of alloying have been proposed as devices tend to be miniaturized in order to provide new materials for the electronic industry with both high strength and high electrical conductivity. And we aim to achieve an optimum combination of super-high electrical conductivity and high tensile strength to set different mechanical processes for treatment as shown in Table 2.

REVIEWER #1:What about the other published works relating to this subject. The authors should add more texts that clearly indicate their point of view with respect to literatures.

Responds:We prefer to retain the existing text for reasons that it has already clearly expressed the reasons for adding V and Co, and using the three different heat treatment methods in the paper, and also explains the effect of alloying of other elements on the properties of Cu-Ni-Si alloys.

REVIEWER #1:The methodology is not clear enough to make the readers of Metals journal can follow this work. More details are required to be added into this section. Schematic figure could be helpful.

Responds:We have provided complete experimental details in Table 2.

REVIEWER #1:Fiqure 4 is not qualitative. The authors have to replace it with high quality image.

Responds:We included a new set of photographs with better definition than those originally submitted.

REVIEWER #1:Figure 8, the shown precipitates are not going well with the direction indicated in the diffraction pattern. Please verify that and supply the good diffraction pattern directions that identify the type of precipitates formed.

Responds:We included a new set of photographs with selected areas of EDS on (001) δ-(Co,Ni)2Si particles in order to better illustrate the composition of the phase.

REVIEWER #1:Figures 10.a, c need to replace with high quality one. The authors should improve the image quality to better clear their point of view.

Responds:We included a new set of photographs with better definition than those originally submitted.

REVIEWER #1:Discussions are not enough. The analysis and interpretation of your results should be improved. Speculative or descriptive discussions are not helpful to reach at a better understanding of the main aim of this work (the point of research). The authors are encouraged to add more explanations with respect to literature. So, add some new references to this section with more explanations.

Responds:As suggested by both referees, a discussion of the possibility of the point of research has been included.(line 251-276)

REVIEWER #1:Conclusions are good but need to be more specific and having the most new points drawn from this work. The conclusions that may add to literature as new points.

Responds:We prefer to retain the existing text for reasons that conclusions have explained the effect of V on the precipitation phase and show an optimum properties combination.

Reviewer 2 Report

New copper alloys, Cu-1.6Ni-1.2Co-0.65Si(-V), were investigated in order to provide  new materials for the electronic industry with both high strength and high electrical conductivity. It was proved that microalloying with vanadium can accelerate the precipitation of solute atoms from the copper matrix in form of d-(Co,Ni)2Si and ß-Ni3Si phases. An optimum combination of super-high electrical conductivity and high tensile strength was achieved for a composition Cu-1.6Ni-1.2Co-0.65Si-0.1V. The prepared material was compared with existing Cu-alloys for similar purposes.

The article is sound and relates the electrical and mechanical properties achieved after different heat treatments to careful microstructure analyses.  

I recommend publication after a few corrections of English grammar (line 44 (and below): founded => found; line 55: focus => focuses; line 105: shorten => shortened; line 105: shorten => shortened).

Author Response

Dear reviewers:

Thank you very much for your careful review and constructive suggestions with regard to our manuscript “Effect of V Addition on Microstructure and Properties of Cu-1.6Ni-1.2Co-0.65Si Alloys” (ID:metals-520883). Those comments are helpful for authors to revise and improve our paper. We have studied comments carefully and tried our best to revise and improve the manuscript and made great changes in the manuscript according to the referees′ good comments. Revised portion is marked in yellow in the paper. The main corrections in the paper and the responds to the reviewer’s comments are as flowing. We appreciate for Reviewers’ warm work earnestly, and hope that the corrections will meet with approval. Please feel free to contact us with any questions and we are looking forward to your consideration. The main corrections in the paper and the responds to the reviewer’s comments are as flowing:

Responds to the reviewer’s comments:

Reviewer #2: (line 44 (and below): founded => found; line 55: focus => focuses; line 105: shorten => shortened; line 105: shorten => shortened).

Responds:line 48,51and 52 (and below): founded => found; line 105: shorten => shortened.

We are very sorry for our negligence of in our manuscript.

Round  2

Reviewer 1 Report

The authors did most of the required modifications. The manuscript can be accepted in its present form.